# A Novel RNA Virus, *Macrobrachium rosenbergii* Golda Virus (MrGV), Linked to Mass Mortalities of the Larval Giant Freshwater Prawn in Bangladesh

**DOI:** 10.3390/v12101120

**Published:** 2020-10-02

**Authors:** Chantelle Hooper, Partho P. Debnath, Sukumar Biswas, Ronny van Aerle, Kelly S. Bateman, Siddhawartha K. Basak, Muhammad M. Rahman, Chadag V. Mohan, H. M. Rakibul Islam, Stuart Ross, Grant D. Stentiford, David Currie, David Bass

**Affiliations:** 1International Centre of Excellence for Aquatic Animal Health, Centre for Environment, Fisheries and Aquaculture Sciences (Cefas), Weymouth, Dorset DT4 8UB, UK; ronny.vanaerle@cefas.co.uk (R.v.A.); kelly.bateman@cefas.co.uk (K.S.B.); stuart.ross@cefas.co.uk (S.R.); grant.stentiford@cefas.co.uk (G.D.S.); david.bass@cefas.co.uk (D.B.); 2WorldFish Bangladesh, Dhaka 1213, Bangladesh; sidhubau1985@yahoo.com (S.K.B.); muhammad.rahman@cgiar.org (M.M.R.); 3Winrock Bangladesh, Dhaka 1212, Bangladesh; sukumar.biswas@winrock.org (S.B.); david.currie@winrock.org (D.C.); 4Centre for Sustainable Aquaculture Futures, University of Exeter, Stocker Road, Exeter EX4 4QY, UK; 5WorldFish, Penang 11960, Malaysia; v.chadag@cgiar.org; 6Bangladesh Fisheries Research Institute, Shrimp Research Station, Bagerhat 9300, Bangladesh; rakib.bfri@gmail.com; 7Department of Life Sciences, the Natural History Museum, London SW7 5BD, UK

**Keywords:** *Macrobrachium rosenbergii*, freshwater prawns, *Nidovirales*, *Macrobrachium rosenbergii* Golda virus, RNA virus, *Roniviridae*, emerging disease, aquaculture

## Abstract

Mass mortalities of the larval stage of the giant freshwater prawn, *Macrobrachium rosenbergii*, have been occurring in Bangladesh since 2011. Mortalities can reach 100% and have resulted in an 80% decline in the number of hatcheries actively producing *M. rosenbergii*. To investigate a causative agent for the mortalities, a disease challenge was carried out using infected material from a hatchery experiencing mortalities. Moribund larvae from the challenge were prepared for metatranscriptomic sequencing. *De novo* virus assembly revealed a 29 kb single-stranded positive-sense RNA virus with similarities in key protein motif sequences to yellow head virus (YHV), an RNA virus that causes mass mortalities in marine shrimp aquaculture, and other viruses in the *Nidovirales* order. Primers were designed against the novel virus and used to screen cDNA from larvae sampled from hatcheries in the South of Bangladesh from two consecutive years. Larvae from all hatcheries screened from both years were positive by PCR for the novel virus, including larvae from a hatchery that at the point of sampling appeared healthy, but later experienced mortalities. These screens suggest that the virus is widespread in *M. rosenbergii* hatchery culture in southern Bangladesh, and that early detection of the virus can be achieved by PCR. The hypothesised protein motifs of Macrobrachium rosenbergii golda virus (MrGV) suggest that it is likely to be a new species within the *Nidovirales* order. Biosecurity measures should be taken in order to mitigate global spread through the movement of post-larvae within and between countries, which has previously been linked to other virus outbreaks in crustacean aquaculture.

## 1. Introduction

The giant river prawn *(Macrobrachium rosenbergii)* is a species of global aquaculture importance, with culture rapidly expanding from <7000 t in 1980 to >237,000 t worldwide in 2018 [1]. Production, particularly in developing parts of Asia, provides direct and indirect employment, a food source to alleviate poverty, and export income in a high value international market [2]. Despite the exponential growth of the prawn industry, disease has been a significant problem, and when mortalities occur they are usually high, causing widespread impacts to the socioeconomic balance in these countries [3].

Bangladesh is considered to be a favourable environment for farming *M. rosenbergii,* named ‘Golda’ by Bangladeshi farmers, due to its vast inland freshwater river systems with adjacent brackish areas and a suitable climate for culturing tropical and sub-tropical aquaculture species [4]. In 2018, Bangladesh was the world’s second largest producer of *M. rosenbergii*, producing 51,571 t [1]. Despite this promising production figure, *M. rosenbergii* hatcheries have been experiencing mass mortalities of larvae, resulting in the number of hatcheries actively producing larvae declining by almost 80% over the past decade as they are unable to complete the production cycle and it becomes uneconomical to continue. Between 2009 and 2010, over 50 hatcheries were operating in Bangladesh, producing up to two hundred million post-larvae (PL) per year; however, in 2019 production had decreased to 27.75 million PL/year from only 12 hatcheries [5,6]. Hatcheries experiencing mortalities reported that larvae had abnormal shapes, reduced appetite, issues with moulting and a gradual whitening of the body, finally resulting in either disappearance from the culture system, or mortality [6].

To compensate for the loss of hatchery-produced post-larvae, wild post-larvae are being caught from the vast river systems in the south of Bangladesh, violating a government ban on the capture of wild PL [4]. Ahmed and Troell [7] reviewed the potential for negative environmental impact associated with fishing for wild PL and identified two main issues; high levels of invertebrate and fish bycatch associated with fishing for larvae (and a subsequent decline in biodiversity) and a reduction in numbers of larvae returning to the freshwater environment as adults to spawn (reducing natural *M. rosenbergii* abundance). As fishing for wild post-larvae increases due to the hatcheries being unable to complete production cycles, it is likely that the negative environmental impact caused by fishing will become more pronounced.

A number of surveys have been carried out by several organisations to determine possible causes for the mortalities observed in hatcheries in Bangladesh. The surveys identified many shortcomings, including deficiencies in water quality, water filtration systems, temperature fluctuations due to weather conditions, feeding practices, probiotic use and poor-quality inputs that had been exposed to formalin and bleaching [5]. However, despite these constraints, hatcheries were successfully completing production cycles until 2011, suggesting that a new factor has arisen, which could be disease-related. Biosecurity has been highlighted as a major issue, in relation to the spread of mortalities throughout the hatcheries of Bangladesh.

As the culture of *M. rosenbergii* has increased globally, the incidence of diseases and emergence of novel pathogens has increased in parallel [3]. Diseases known to affect different life stages of the giant freshwater prawn include numerous pathogenic bacteria, viruses and fungi. Opportunistic bacteria such as *Aeromonas* spp., *Pseudomonas* spp. and *Vibrio* spp. can cause infections in all life stages of *M. rosenbergii* culture (reviewed in Pillai and Bonami [3]). *Spiroplasma eriocheiris* [8] has been shown to cause mortalities in later life stages of *M. rosenbergii.* Macrobrachium rosenbergii Nodavirus (MrNV) [9], in association with extra small virus (XSV) [10], causes white tail disease (WTD), a disease listed by the World Organisation for Animal Health (OIE). The viruses have caused devastation in the prawn hatchery industries of Thailand [11], China [10], India [12], Taiwan [13] and Indonesia [14]. Other viruses causing mortalities include Macrobrachium rosenbergii Taihu Virus (MrTV), a novel dicistrovirus associated with larval mortalities in China [15], Infectious Hypodermal and Haematopoietic Necrosis Virus (IHHNV) [16], and Decapod Iridescent Virus (DIV1) [17]. Other pathogens have been identified to infect *M. rosenbergii* without large mortalities including: Macrobrachium Parvo-like Virus (MPV) [18] and White Spot Syndrome Virus (WSSV) [19,20]. With the lack of immortal shrimp cell lines, disease challenges can be used to identify pathogenic agents infecting *M. rosenbergii*. This technique was recently used to determine that MrNV alone can cause mortalities in *M. rosenbergii* in the absence of XSV [21].

Alam et al. (2019) attempted to determine the causative agent(s) associated with the larval mortalities in Bangladesh hatcheries using a screen for *Vibrio* spp., which are known to cause disease in larval stages of prawn culture [22]. However, this study was unable to find a strong correlation between the *Vibrio* species present and the levels of mortalities seen in larvae. Larvae were also screened for the presence of MrNV and XSV, but neither of these were detected. Therefore, no significant pathogens have yet been associated with the large-scale larval mortalities in Bangladesh. The clinical signs of disease observed in these larvae associated with the current large-scale mortalities do not precisely fit those described for infection with any of the known larval prawn pathogens; therefore, an approach to detect novel pathogens was needed to determine if there was a pathogenic cause for the mortalities.

In this study we (1) carried out an experimental challenge trial using larvae sourced from a hatchery displaying disease, (2) generated and analysed metatranscriptomic libraries obtained from pools of larvae from experimental trials to identify any potential pathogens present in moribund animals, and (3) screened for pathogens identified as a risk in other hatcheries that had been experiencing mortalities to determine whether these agents were present.

## 2. Materials and Methods 

### 2.1. Collection of Hatchery Cultured Larvae and Wild Broodstock

Larvae were collected from six hatcheries in southern Bangladesh over the production seasons of 2018 and 2019. Batches of approximately 20–30 whole larvae were fixed in RNAlater. In the same years, adult *M. rosenbergii* were collected from seven rivers, also in southern Bangladesh (Figure 1). 

### 2.2. Preparation of Infectious Study Material

To produce materials for disease challenge, moribund larvae were collected from a hatchery that was experiencing mortalities at the time of the study. Larvae were ground using a sterile pestle and mortar, suspended in phosphate buffered saline (PBS) and filtered through a 0.45 µm syringe filter. Larvae from a healthy hatchery that had not experienced mortalities in past production cycles were also collected and prepared in the same manner as a control treatment. 

### 2.3. In Vivo Tank Experiments

Experimental larvae were obtained from Bangladesh Fisheries Research Institute’s (BRFI) domesticated F1 broodstock. After hatching, larvae were reared up to stage three, before splitting into three groups for different experimental tank exposures.

Experimental larvae were directly immersed into filtered challenge medium from moribund larvae for 10 min prior to being transferred to a fibreglass tank.Experimental larvae were fed artemia that had been immersed in challenge medium from moribund larvae for 10 min.Experimental larvae were immersed into filtered challenge medium from healthy larvae for 10 min prior to being transferred to a fibreglass tank.

For all experiments, 4000 ± 200 larvae were used, and experiments were carried out in 40 L fibreglass tanks heated to a constant temperature of 30 °C with a submerged thermostat heater. Larvae were fed three times per day (25 g/1000 L) with Red Jungle Brand^®^ Artemia (Ocean Star International, Snowville, UT, USA). Twenty to thirty larvae were collected every day for the first five days of the experiment and on days seven and ten.

### 2.4. RNA Extraction

RNA was extracted using RIBOZOL™ (VWR, Radnor, PA, USA). For larvae, approximately 20 larvae were added to 1 mL RIBOZOL, and for adults, approximately 5 mg of each tissue type (pleopod, hepatopancreas, gill, gut) was pooled and added to 1 mL of RIBOZOL. RNA extraction was carried out following the manufacturer’s protocol, with a final resuspension in 50 μL molecular grade water (Fisher Scientific, Waltham MA, USA). To remove any DNA co-extracted with RNA, a DNAse step was carried out using DNAse1 (Sigma Aldrich, St. Louis, MO, USA), following the manufacturer’s protocol.

### 2.5. Double Stranded cDNA Synthesis

First strand synthesis was performed in 25 μL reaction volumes using 5 μL of MMLV-RT 5× buffer (Promega, Madison, WI, USA), 1.25 μL of dNTPs (10 mM) (Promega), 0.625 μL of recombinant RNasin (Promega), 1 μL of random hexamer primers (Promega) and 1 μg of RNA in 16.125 μL. Prior to the addition of 1 μL of MMLV-RT enzyme (Promega), the reaction was incubated at 65 °C in order to denature double stranded RNA. Following the addition of the MMLV-RT, the reaction was incubated at 37 °C for 1 h. Second strand synthesis was performed immediately after first strand synthesis. Single-stranded cDNA was incubated at 94 °C for 2 min, followed by 10 °C for 5 min. Following incubation, 2 μL Sequenase buffer (Thermo Fisher, Waltham, MA, USA), 0.3 μL Sequenase (Thermo Fisher) and 7.7 μL molecular grade water (Thermo Fisher) were added. Following this, reactions were incubated at 37 °C for 8 min, 94 °C for 2 min and 10 °C for 5 min. During incubation at 10 °C, 1.2 μL 1:4 (Sequenase:Sequenase dilution buffer (Thermo Fisher)) diluted Sequenase was added to reactions and incubated at 37 °C for 8 min and 94 °C for 8 min prior to transfer to ice.

### 2.6. Library Preparation

Three pools of cDNA were constructed from in vivo tank experiments: (1) Challenge days 1–4 larvae from immersion and feeding challenges using filtered medium from moribund larvae (2) Challenge days 5–10 larvae from immersion and feeding challenges using filtered medium from moribund larvae and (3) Larvae from a hatchery that had been experiencing mass mortalities. These pools were prepared for metatranscriptomic sequencing using the Illumina compatible Nextera XT library preparation kit (Illumina, San Diego, CA, USA) and sequenced on an Illumina MiSeq using v3 chemistry (Illumina).

### 2.7. Sequence Analysis 

The generated raw Illumina paired-end sequence reads were trimmed to remove adaptor and low quality sequences using Trim galore! v0.5.0 [23]. Trimmed sequences were quality-checked using FastQC v0.11.8 [24] before reads from individual pools were assembled using both rnaSPAdes v3.13.0 [25] and the Iterative Virus Assembler (IVA) v1.0.8 [26]. Assembled contigs were subsequently annotated using the BLASTp algorithm of Diamond v3.13.0 [27] and the full NCBI non-redundant (nr) protein database (downloaded November 2018), and the results were visualised using MEGAN6 Community Edition v6.12.3 [28]. Paired reads from all samples were then mapped to the assembled contig representing the viral genome sequence using BWA-MEM v0.7.17 and SAMtools v1.9 with default parameters [29,30]. The output from BWA-MEM was visualised with Integrative Genomics Viewer (IGV) v2.5.2 [31]. Assembly quality and accuracy were assessed with QualiMap v2.2.2 [32]. 

Predicted open reading frames (ORFs) were identified using four different tools: Prokka v1.11 [33] FgenesV0 (SoftBerry.com), GeneMarkS v4.28 [34] and Vgas [35]. ORFs that were supported by two or more programs were analysed further. Supported ORFs were annotated using NCBI BLASTp and the full NCBI protein sequence database (downloaded December 2019) and protein motifs were identified by HHpred [36] (default parameters) and InterProScan 5 [37]. Predicted protein motifs were aligned against known nidovirus protein motifs using MAFFT [38] and multiple sequence alignments (MSAs) were visualised in ESPRICT3 v3.0 [39] (default parameters). Transmembrane regions were identified using TMHMM v2.0 [40] (default parameters). Ribosomal frameshift identifiers were found using FSFinder2 using the virus genome settings [41]. Secondary structure of the 3′ untranslated region (UTR) was predicted using MFOLD [42] (default parameters).

### 2.8. Phylogenetic Analyses

Phylogenetic analyses was performed using the RNA-dependent RNA polymerase (RdRp) protein domain as in Saberi et al. [43], conserved in known and proposed *Nidovirales,* on 15 *Nidovirales* with representatives from *Arteriviradae*, *Roniviridae, Mesoniviridae, Mononiviridae, Euroniviridae*, *Coronavirinae* and *Torovirinae* families and subfamilies. Two representatives from the *Astroviradae* order of viruses were also aligned as an outgroup. MSAs were performed using default parameters in MAFFT [38]. Maximum likelihood phylogenetic analyses were carried out using RaxML BlackBox v.8 [44] (Generalised time-reversible (GTR) evolutionary model; all parameters estimated from the data). A Bayesian consensus tree was constructed using MrBayes v.3.2.6 [45]. Two separate Multi-Core Markov-Chain Monte Carlo (MC3) runs with randomly generated starting trees were carried out for 2 million generations each with one cold and three heated chains using a GTR model. All parameters were estimated from the data. The trees were sampled every 1000 generations and the first 500,000 generations discarded as burn-in. All phylogenetic analyses were carried out on the Cipres server [46].

### 2.9. Primer Design and Specific PCR

Primers specific to the predicted enveloping protein ORF of MrGV were designed using Primer3 v4.1.0 [47] using default settings and an amplicon length of 250–400 bp: MrGV_F1: 5′-TTTGCCCAGGTTAATTGCCC-3′ and MrGV_F2: 5′-ACAAGTGCCAGTGAGACGTA-3′, producing an amplicon of 319 bp. PCR amplification was performed in 25 μL reactions using 5 μL 5× Green GoTaq Flexi Buffer (Promega), 1.5 μL MgCl_2_ (25 mM) (Promega), 0.2 μL of each primer (10 μM), 0.5 μL dNTPs (25 mM) (Promega), 0.125 μL GoTaq polymerase (Promega), 15.925 μL molecular grade water and 1.25 μL of template cDNA. Initial denaturation was carried out at 95 °C for 5 min, followed by 30 cycles of 95 °C for 30 s, 58 °C for 30 s and 72 °C for 30 s. This was followed by a final extension at 72 °C for 10 min. Amplicons were purified with Wizard^®^ SV Gel and PCR Clean-Up System (Promega) and sequenced via the Eurofins TubeSeq service. Primers were tested on larval cDNA from five hatcheries and cDNA from wild adult *M. rosenbergii* sampled from the river networks surrounding the hatcheries. cDNA from *Penaeus monodon* tissues infected with yellow head virus (YHV) were tested as a negative control.

### 2.10. Specific PCR Screens

PCR screens for specific pathogens were carried out with the following primers: MrNV using MrNV2aF and MrNV2aR primers [48]; XSV using XSV-external forward and reverse primers [49]; MrTV using MrTV472F and MrTV472R primers [15]; WSSV using 146F1, 146R1, 146F2 and 146R2 primers [50]; Penaeus monodon nudivirus (PmNV) using 261F and 261R primers [51]; *Spiroplasma eriocheiris* using F28 and R5 primers [52]; and YHV using YC-F1a, YC-F1b, YC-R1a, YC-R1b, YC-F2a, YC-F2b, YC-R2a and YC-R2b primers [53]. All PCR reactions were performed as above using the cycling conditions specified in the original publications.

## 3. Results

### 3.1. In Vivo Tank Experiments

At the end of the 10-day disease challenge, mortalities exceeding 80% had occurred in all three experimental groups, including larvae exposed to an extract produced from a hatchery that at the time had not experienced mortalities (Figure 2). However, it was later discovered that the healthy hatchery experienced mass mortalities to a similar level of other hatcheries in the south of Bangladesh shortly after they were sampled for the experimental material. As the challenge progressed, reduction in swimming ability, feeding and growth were observed in all treatments. Moribund larvae appeared white in colour compared to healthy larvae.

### 3.2. Complete Genome Assembly of Virus

A total of 3,466,390, 3,556,054 and 5,888,915 Illumina read-pairs were generated for libraries 1, 2 and 3, respectively, and after quality-trimming and filtering, 3,453,843, 3,545,471 and 5,867,622 read-pairs remained. Contigs assembled using rnaSPAdes, both separately and by combining all three libraries, were annotated using Diamond in BLASTx mode. rnaSPAdes assembly of combined libraries produced 38,826 contigs; 23 contigs, of average length 4560 bp, had similarity in protein sequence to YHV or gill-associated virus (GAV), but when the trimmed reads were aligned against the YHV genome (accession number GCA_003972805.1), no alignment was seen. De novo virus assembly using IVA from library three produced 12 contigs, and two non-overlapping contigs, of lengths 14,858 and 13,463 had BLASTx similarity to YHV. IVA assembly using a pool of all three libraries produced a full genome consensus sequence of 29.11 kb, with an average coverage of 1630×. The two contigs produced by IVA assembly of library three mapped to the 29.11 kb consensus sequence. The full genome sequence is deposited under accession number MT907511 on GenBank. Henceforth, we refer to this novel virus as Macrobrachium rosenbergii Golda Virus (MrGV). Mapping trimmed and quality-filtered reads from each individual library back to the consensus genome gave coverage of 5.37× for library one (pooled challenged larvae from days 1–4), 269.28× for library two (pooled challenged larvae from days 5–10) and 1462.64× coverage for library three (pooled larvae from a hatchery experiencing mortalities). To compare the efficiency of the rnaSPAdes assembly, assembled contigs from individual and pooled libraries were mapped back to the consensus genome. Eleven contigs from library one mapped to MrGV with 0.25× coverage, 32 contigs from library two mapped with 2.53× coverage, 50 contigs from library three mapped with 5.04× coverage and 81 contigs from the pooled libraries mapped with 7.25× coverage. rnaSPAdes contigs from libraries two and three gave good coverage over the whole genome but failed to assemble the 5′ and 3′ ends. A comprehensive table of assembly statistics is in Appendix A.

### 3.3. Open Reading Frame (ORF) Prediction, Genome Prediction and Motif Identification

FgenesV0 predicted five protein-coding ORFs in the positive sense direction, four of which were supported by GeneMarkS, Prokka and Vgas (Figure 3), which were investigated in more detail. The two longest ORFs, ORF1a and ORF1b, showed homology to the replicase polyproteins of yellow head virus (Evalues of 1×10^−29^ and 0), whereas ORF3 showed homology to glycoproteins associated with species in the Negarnaviricota phylum (realm Riboviria), including a number of species in the genus *Orthobunyavirus* (*E* value of 1×10^−14^). ORF2 had no significant similarity to any known proteins. HHpred produced confident predictions (≥99%) for a picornain-like protease and endopeptidase enzyme in ORF1a, a coronavirus-like RNA-dependent RNA polymerase (RpRp), a metal-binding helicase and a 3′-5′ exoribonuclease (ExoN) in ORF1b. HHpred did not detect any zinc-binding domains (ZBDs); however, when ORF1b was run against the InterPro database using InterProScan5, a coronavirus-specific ZBD was identified. In ORF3, HHpred identified an enveloping glycoprotein associated primarily with viruses of the order Bunyavirales. Both HHpred and InterProScan5 were unable to identify any other protein domains; therefore, predicted MrGV ORFs were aligned against the nidovirus domains identified in Saberi et al. [43]. MSA identified a further three protein motifs: A nidovirus RdRp-associated nucleotidyltransferase (NiRAN) and S-adenosylmethionine (SAM)-dependent N7- and 2′-O-methyltransferases (N-MT and O-MT, respectively).

### 3.4. Identification of Frameshift Motifs

As *Nidovirales* typically translate ORF1a and ORF1b consecutively to produce pp1ab by −1 ribosomal frame shift [54], FSFinder2 [41] was used to identify −1 frameshift sites in the overlap between ORF1a and ORF1b. Both major elements of −1 frameshifting were identified in the overlap: A slippery site at position 15,090 nt with the sequence “GGGTTTT”, proceeded by a stem-loop stimulatory structure located a few nucleotides downstream between positions 15,107 and 15,165 nt.

### 3.5. Prediction of 3′ UTR Secondary Structure

Analysis of the 3′ UTR secondary structure following the final stop codon of the final 3′ ORF revealed a thermodynamically stable RNA hairpin secondary structure (Figure 4). The predicted hairpin structure of MrGV is stabilised by three helices with a 12 nt hairpin loop (ΔG = −40.5), in comparison to the GAV 3′-UTR structure, which forms a 8 nt hairpin loop stabilised by four helices (ΔG = −32.40), previously identified in Wijegoonawardane et al. [55].

### 3.6. Phylogenetic Characterisation of MrGV

A Bayesian consensus tree was produced based on the RdRp protein domain universally conserved in nidoviruses, including the RdRp sequence generated in this study for MrGV, and the only nidovirus protein domain able to align to the RdRp of *Astrovirales* (outgroup). All families within *Nidovirales* were monophyletic in our analyses. MrGV branched as sister to GAV and YHV within the *Roniviridae* clade (Figure 5). Other invertebrate-infecting *Nidovirales* shown in Figure 5 include Charynivirus-1 infecting the crab *Charybdis*, and Paguronivirus-1 infecting hermit crabs [56], all branching within the *Euronivirdivae* clade. All invertebrate-infecting nidoviruses branched together with maximal Bayesian Posterior Probability.

### 3.7. Multiple Sequence Alignment (MSA) of 3′ ORFs

As MrGV branches within *Roniviridae*, which have a distinct 3′ genome organisation to other nidoviruses [57], MrGV ORF2 and ORF3 were aligned against the corresponding ORFs of GAV. This alignment (Figure 6) showed similarities in protein sequences of the three 3′ transmembrane helices of ORF3 of GAV to the predicted transmembrane helices of MrGV. Sequence similarities were also seen in ORF2 of GAV to ORF2 of MrGV (18.24% residue similarity), as well as gp116 (15.40% residue similarity) and gp64 (17.03% residue similarity) protein coding regions of ORF3 in comparison to the aligned regions in MrGV ORF3.

### 3.8. Infection Prevalence in Hatchery-Reared Larvae and Wild River-Based Adult Populations

The virus was not detected by specific PCR in cDNA from any adult tissues from wild river populations across the two-year sampling period (119 animals) and *S. eriocheiris* was the only pathogen detected in adult *M. rosenbergii* (10/119). However, MrGV was detected in larval cDNA from all three hatcheries sampled in 2018 production cycles and all five hatcheries sampled in 2019 production cycles with no other pathogens detected by PCR in hatchery larvae. MrGV was also detected in BRFI ‘control’ hatchery larvae prior to challenge, potentially explaining the low survival of Treatment 3 larvae. No histopathological signs of infection were seen in the larvae.

## 4. Discussion

We report and analyse the complete genome of a novel single-stranded positive-sense RNA virus infecting cultured *M. rosenbergii* from hatcheries in southern Bangladesh. The novel virus, Macrobrachium rosenbergii Golda virus, has a similar genome arrangement to viruses of the order Nidovirales, and appears to be most closely phylogenetically related to yellow head virus and gill-associated virus, both infecting penaeid shrimp.

MrGV was detected by specific PCR in larvae from three hatcheries in 2018, two of which were experiencing mass mortality events when sampled, and one which was sampled just prior to a mass mortality event. In 2019, two of the hatcheries that had experienced mass mortalities the previous year were re-sampled during mass mortality events and were again positive for MrGV; both of these hatcheries underwent two production cycles, both suffering mass mortality events and both were PCR positive for MrGV. In addition to the two hatcheries that were re-sampled in 2019, a further three hatcheries, none of which had been sampled in 2018, were sampled in 2019 during mass mortality events. These further three hatcheries were also PCR positive for MrGV. Given the number of hatcheries affected by mass mortalities linked to MrGV, its temporal prevalence and spatial spread, with one of the hatcheries over 40 km in distance to the nearest hatchery that was PCR positive for MrGV, we suggest that this novel virus represents a very significant threat to *M. rosenbergii* aquaculture within Bangladesh, and may be a significant factor in the collapse of larval production in the industry of Bangladesh since 2010.

All PCR screens of larvae collected from hatcheries experiencing mass mortalities for pathogens known to infect the larval stage of *M. rosenbergii* were negative: MrNV and XSV, the causative agents of white tail disease [9,10]; MrTV, a virus associated with mass larval mortalities in China [15], *Spiroplasma eriocheiris* [8], and WSSV—shown to be able to infect *M. rosenbergii* experimentally [19]. Larvae were also screened for YHV and PmNV, viruses known to infect all life stages of marine shrimp species [58,59]; these screens were all negative. Furthermore, no sequences were assignable to any other known pathogens of *M. rosenbergii* in our metatranscriptomic data. The absence of these pathogens and no obvious bacterial cause observed by Alam et al. [6] strongly suggests that the survival problems in Bangladesh hatcheries are not due to a currently known pathogen and that mass mortalities were linked either to hatchery practice factors or/and the emergence of a novel pathogenic agent. Despite numerous problems identified in hatchery practices, hatcheries were successfully producing until 2011, and since then, hatchery practices have not changed, suggesting that this is not the main source of mortality events. PCR screens of adult *M. rosenbergii* cDNA from rivers used to collect berried females as the supply of broodstock for the hatcheries were also all negative for MrGV, suggesting that broodstock may not be the entry route of MrGV into the hatcheries. Adults were also negative for all other pathogens as above, except for *S. eriocheiris*. All larvae sampled from hatcheries experiencing mortalities were negative for *S. eriocheiris* and no reads for any *Spiroplasma* species were detected in metatranscriptomic analysis of the moribund larvae, suggesting that *S. eriocheiris* is also not causing the mortalities in hatcheries.

Nidoviruses (order *Nidovirales*) are enveloped positive-sense RNA viruses infecting a range of hosts including both vertebrates and invertebrates [54]. The invertebrate nidoviruses are composed of families: *Mesoniviridae, Roniviridae, Mononiviridae* and *Euroniviridae*, with the latter two families discovered within the last four years [43,54,56]. *Roniviridae* comprises one genus, *Okavirus*, composed of two closely related crustacean-infecting viruses: GAV and YHV [54]. Both GAV and YHV are associated with disease in marine shrimp farming, with the former initially associated with mid-crop mortality syndrome (MCMS) in *Penaeus monodon* in Australia [60] and the latter first associated with yellow head disease (YHD) in *P. monodon* in Thailand [61]. Prior to this study, no nidoviruses had been discovered in *Macrobrachium rosenbergii.*

When amino acid sequences of MrGV were screened against the NCBI non-redundant protein database, there was weak similarity to YHV and even weaker similarity to other nidoviruses. However, the only significant matches in pp1a and pp1ab were to this order; therefore, we searched for protein motifs shared among other nidoviruses. Through database searches we identified five protein motifs present in all known nidoviruses: A protease, an RNA-dependent RNA polymerase, a zinc-binding domain, a helicase and a methyltransferase-exoribonuclease complex. A further three domains were found by sequence alignment of MrGV to the protein motifs of NiRAN and (SAM)-dependent N7- and 2′-O-methyltransferases (N-MT and O-MT, respectively). We also identified nucleotide sequences within the overlap of ORF1a and ORF1b suggestive of a −1 ribosomal frameshift. The slippery sequence “GGGTTTT” proceeded downstream by a putative stem-loop stimulatory RNA structure suggests that pp1a and pp1ab are translated continuously by programmed ribosomal frameshift, a characteristic of nidoviruses [62]. The genome of *Okavirus*, the genus to which GAV and YHV belong, have a unique genome architecture to that of vertebrate nidoviruses—ORF2, encoding a nucleocapsid protein, is located upstream of the glycoprotein gene (ORF3) [57]. This structure is also seen in MrGV, as well as sequence similarity of ORF2 and ORF3 to the corresponding ORFs in GAV.

A characteristic of positive-sense RNA viruses is the presence of a secondary RNA structure at the 3′ UTR; this is present in other members of the nidoviruses including *Coronavirdae* and *Arteriviradae* [63]. These 3′ UTR structures and/or specific sequences have been shown to be critical to polymerase recognition and minus-strand genomic RNA synthesis [64]. A 3′ UTR secondary structure has been identified in GAV and YHV, and appears to be well conserved, with complementary base-changes implemented in YHV genotypes to retain a conserved stem-loop structure stabilised by four helices [55]. In this study we used MFOLD to predict the secondary RNA structure of MrGV in the 3′ UTR of the genome following the final stop codon of the 3′ ORFs. MFOLD predicted a stem loop structure of a similar size to that of GAV and YHV, with three helices stabilising a 12 nt hairpin loop. It is hypothesised that the 3′ UTR RNA structure in GAV and YHV may act as a polymerase recognition site for minus-strand RNA synthesis [55], and given the similarities in the 3′ UTR RNA structure in MrGV to GAV and YHV, this structure in MrGV may have a similar function. 

Based on multiple sequence alignment of the RdRp protein motifs of a representation of families within *Nidovirales*, MrGV groups phylogenetically with family *Roniviridae*, which, thus far, exclusively comprises GAV and YHV. We hypothesise that MrGV also belongs to *Roniviridae*; however, further sampling is needed to obtain additional material for electron microscopy to visualise the virus in situ in order to confirm that virion morphology conforms with the rod-shaped characteristics of the family [54].

Our study has also highlighted that choice of assembly tool is important in assembling viruses. rnaSPAdes was not able to assemble the complete MrGV genome, even when libraries were pooled to give higher coverage than single libraries alone, whereas IVA was able to assemble the full genome from the same pool of libraries. It is likely that IVA was able to assemble the full genome as it is capable of assembling RNA sequences at highly variable depths [26], a useful tool in the case of the nidoviruses, which are prone to variability in sequencing depth due to the production of sub-genomic mRNAs (sgmRNAs). The presence of sgmRNAs were not assessed in this study but given that rnaSPAdes could not assemble the full genome, it is likely that these sgmRNAs do exist. Future work includes identifying the presence or absence of sgmRNAs and potential transcription regulation sites (TRSs).

Samples for histology or electron microscopy matching those used for molecular analyses were not available from the disease challenge studies carried out here. Therefore, there is currently no information about histopathological signs of disease caused by MrGV. Histological samples were taken from the larvae sampled from hatcheries in 2019, however these animals were likely not to be moribund and were probably in the early stages of infection by MrGV and thus, though infected, no pathology was seen. Furthermore, RNA viruses can only be visualised indirectly in histology sections (normally associated with damaged host cells and nuclei). Further sampling is ongoing in order to visualise the virus infecting larval tissue(s) using transmission electron microscopy.

The challenge experiment data from this study were unable to determine the route of entry of MrGV into the hatchery system, as it was later determined that control animals were sourced from a hatchery assumed to be free of MrGV, but which after sampling larvae for the challenge experienced mass mortalities to a similar level as seen in other hatcheries in southern Bangladesh. Larvae were sampled from the apparently unaffected hatchery and fixed before challenging; these animals were screened by PCR for all pathogens as above and were only positive for MrGV, thus suggesting that animals were infected with MrGV prior to the challenge, explaining the high levels of mortality in ‘control’ animals. This confounding situation nonetheless further suggests the role and ubiquity of MrGV in mass mortality events across multiple hatcheries in Bangladesh. Future work to identify how the virus is entering the hatcheries is ongoing to suggest preventative methods that could be implemented to ensure biosecurity. 

The use of metagenomic techniques to identify both novel RNA and DNA viruses is becoming more common in aquaculture, with new viruses identified in economically-important species including fish, crustaceans and molluscs (reviewed in Munang’andu et al. [65]). Metatranscriptomics has recently been used to identify another novel RNA virus in *M. rosenbergii,* Crustacea hepe-like virus 1 (CHEV1), associated with animals exhibiting growth retardation in China [66]. The identification of MrGV is the first study, to our knowledge, to use a metatransciptomic approach to investigate the mass mortalities experienced in hatchery-reared prawn larvae in Bangladesh. Previous studies have investigated possible agents involved in mortality events using PCR screens and microbial culture, but required the agent to have gene sequences sufficiently similar to known pathogens to amplify with PCR primers or culturable on media, respectively, thus limiting the detection of more genetically divergent pathogens, possibly including many viruses. The metatranscriptomic approach used in this study to discover the agent involved in the Bangladesh mortalities was able to identify and characterise a novel pathogen that would likely have not been identified by most commonly employed methods. Furthermore, this method can be used to rapidly detect and comprehensively analyse new and cryptic viral linages, of relevance to both animal and human health.

## Figures and Tables

**Figure 1 viruses-12-01120-f001:**
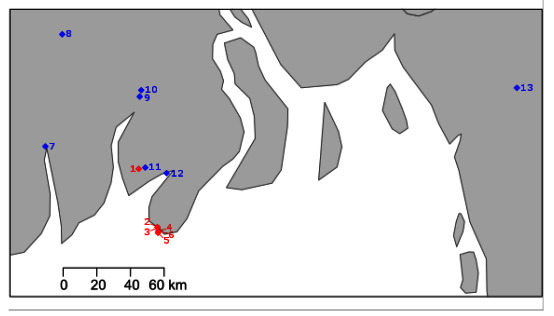
Map of the southern Bangladesh coastline (sea in white, land in grey) with sampling site locations. Hatcheries denoted with a red diamond (1–6) and river sampling sites denoted with blue diamond (7: Shibsa River, 8: Rupsa River, 9: Katcha River, 10: Kocha River, 11: Biskhali River, 12: Payra River, 13: Karnaphuli River).

**Figure 2 viruses-12-01120-f002:**
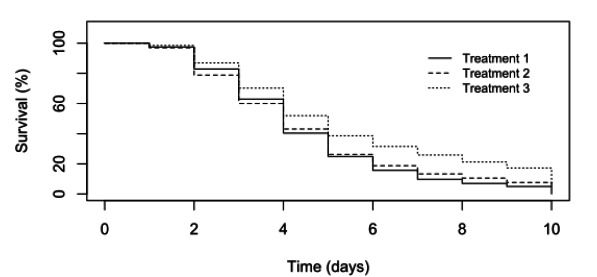
Larval mortalities observed in tank experiments. Treatment 1 = Larvae directly immersed into filtered material from moribund larvae. Treatment 2 = Larvae fed artemia that had been immersed into the filtered material from moribund larvae. Treatment 3 = Experimental larvae were directly immersed into filtered material from healthy larvae.

**Figure 3 viruses-12-01120-f003:**
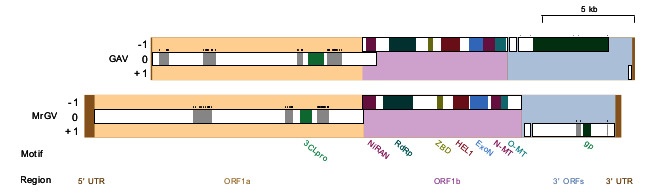
Schematic illustration of linear Macrobrachium rosenbergii Golda Virus (MrGV) and gill-associated virus (GAV) genomes and proteomes. Open reading frame (ORF) 1a is set as reading frame zero and genomes are spit into five sections: 5′ untranslated region (UTR), ORF1a, ORF1b, 3′ ORFs and 3′ UTR. Transmembrane (TM) regions are shown in grey with predicted TM helices shown as black bars above these regions. Predicted protein motifs are a 3C-like protease (3CLpro), nidovirus RdRp-associated nucleotidyltransferase (NiRAN), RNA-dependent RNA polymerase (RdRp), zinc-binding domain (ZBD), superfamily 1 helicase (HEL1), 3′-5′ exoribonuclease (ExoN), S-adenosylmethionine (SAM)-dependent N7- and 2′-O-methyltransferases (N-MT and O-MT, respectively) and glycoproteins (gp).

**Figure 4 viruses-12-01120-f004:**
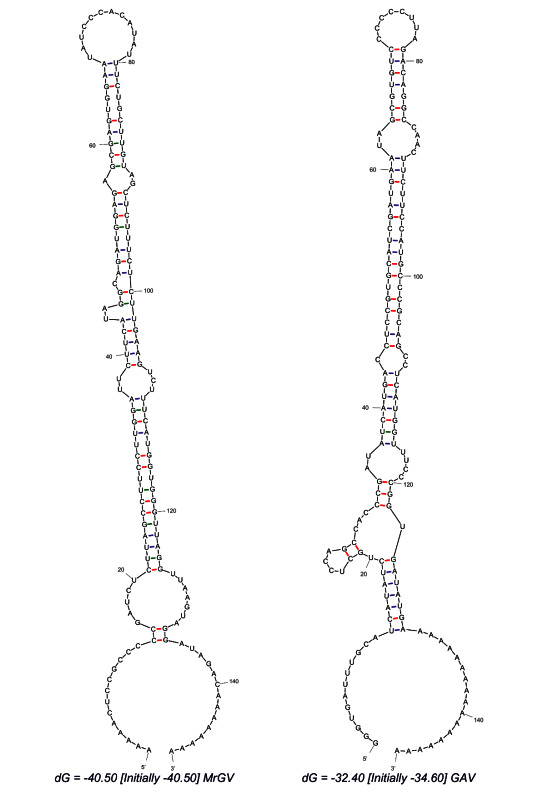
RNA secondary structure predicted using MFOLD of 3′ UTR sequences of MrGV and GAV downstream of the stop codon of the final 3′ ORF.

**Figure 5 viruses-12-01120-f005:**
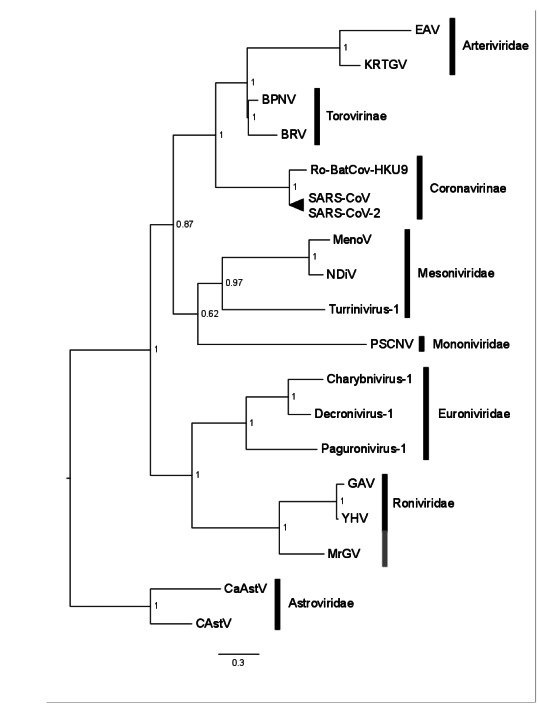
Bayesian consensus tree based on the RdRp of a representative set of nidoviruses, MrGV and astroviruses (outgroup). Accession numbers and names in Appendix A.

**Figure 6 viruses-12-01120-f006:**
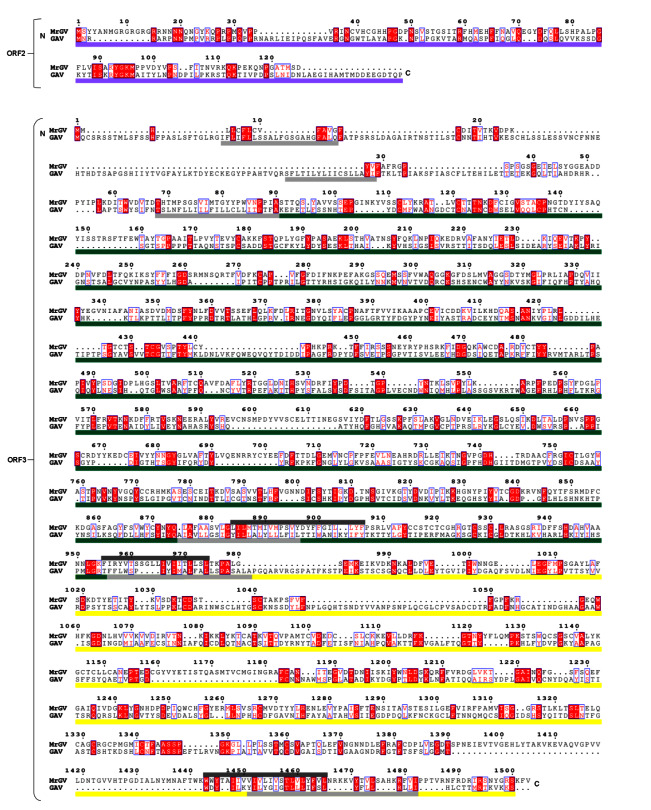
Sequence alignment of MrGV and GAV 3′ UTRs. Residue numbering based on the start codon of MrGV ORFs. ORF2 of GAV encodes the p20 nucleoprotein (purple bar). ORF3 of GAV encodes the gp116 glycoprotein (green bar) and gp64 glycoprotein (yellow bar). Transmembrane regions of GAV are shown as light grey bars and TMHMM-predicted transmembrane regions of MrGV are shown are dark grey bars. Residues in red boxes are conserved and residues in blue boxes have >0.7 Global Similarity Score based on physio-chemical similarities in ESPript3.

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
