# Peer review of "A Novel RNA Virus, Macrobrachium rosenbergii Golda Virus (MrGV), Linked to Mass Mortalities of the Larval Giant Freshwater Prawn in Bangladesh"

_viruses, 2020, doi:10.3390/v12101120_

Round 1

Reviewer 1 Report

The bioinformatics in this article is thorough and well written, I have very few suggestions regarding this area.  The couple I have are as follows; The use of metatranscriptomics is an emerging tool to identify RNA viruses in aquaculture environments.  It has been done elsewhere in the world & would be reasonable to acknowledge this in the discussion, probably when you describe this as being the first time it has been carried out in Bangladesh.  There is also a typo on line 425 where a space needs to be added between words.

I have several comments and suggestions regarding other aspects of the paper.  Please see below.

Figure 1:  It is not clear what the grey and white represent – coastline?  Please describe in figure title.  It might be helpful to identify whether any hatcheries are upstream/near river sites.  Is it possible for this disease to spread from one to the other (water flow/bird vector)? This is  a minor query that may be outside your papers scope.

In vivo tank experiments

It is unclear as to whether the BRFI larvae used in this experiment were clear of the virus prior to experimentation.  A control group with no exposure to challenge media of any kind would be useful to identify the amount of mortality due to challenge & what is natural attrition or other causes.  Although this is unlikely to be something you can do now, PCR confirmation of no virus in the BRFI larvae would substitute to clarify results.   If you can access this (using kept samples), or qualify the results with the information that you do not have access to this, it would strengthen this area of the paper.  Can you also confirm in the paper what the larvae were fed during the experiment?  Type, quality and origin of food could also be a route of infection.

Author Response

The bioinformatics in this article is thorough and well written, I have very few suggestions regarding this area.  The couple I have are as follows; The use of metatranscriptomics is an emerging tool to identify RNA viruses in aquaculture environments.  It has been done elsewhere in the world & would be reasonable to acknowledge this in the discussion, probably when you describe this as being the first time it has been carried out in Bangladesh. 

>> Cited a recent review of viral discovery in aquaculture that gives examples of viruses discovered using metagenomic tools, and added a recent examples of metatranscriptomics being used to identify another novel RNA virus in M. rosenbergii.

There is also a typo on line 425 where a space needs to be added between words.

>> Typo on line 425 corrected from "obtainadditional" to "obtain additional".

I have several comments and suggestions regarding other aspects of the paper.  Please see below.

Figure 1:  It is not clear what the grey and white represent – coastline?  Please describe in figure title.  It might be helpful to identify whether any hatcheries are upstream/near river sites.  Is it possible for this disease to spread from one to the other (water flow/bird vector)? This is a minor query that may be outside your papers scope.

>> Revised figure 1 legend to clarify that it is a map of the coastline of Bangladesh with land mass in grey.

In vivo tank experiments

It is unclear as to whether the BRFI larvae used in this experiment were clear of the virus prior to experimentation.  A control group with no exposure to challenge media of any kind would be useful to identify the amount of mortality due to challenge & what is natural attrition or other causes.  Although this is unlikely to be something you can do now, PCR confirmation of no virus in the BRFI larvae would substitute to clarify results.   If you can access this (using kept samples), or qualify the results with the information that you do not have access to this, it would strengthen this area of the paper. 

>> Clarified that BRFI larvae were positive for MrGV prior to challenge in section 3.8 (lines 348-350). This is also discussed in lines 453-463 of the discussion.

Can you also confirm in the paper what the larvae were fed during the experiment?  Type, quality and origin of food could also be a route of infection.

>> Added brand of artemia and amount fed to larvae during tank experiments in section 2.3

Reviewer 2 Report

This study reports mainly on the outcome of a de novo virus assembly that revealed a novel 29 kb single stranded positive-sense RNA virus that is the likely etiological agent of  mass mortalities in the giant freshwater prawn, Macrobrachium rosenbergii.  

The paper is well written and data is clearly presented.  I do have one major concern.  The wider screening activity (sections 2.9 & 3.8) appears to be flawed as this virus in question is a RNA virus without a DNA life stage.  So PCR in itself won't work.  It is unclear from the methods section whether the samples used in the wider screen was in fact cDNA.  I'm hoping that this is just a misunderstanding, and if so this must be corrected or clarified.

Similarly, I recommend the authors use the term RT-PCR rather than PCR when reporting on the screening results.  

I noticed one minor typo: line 243 - this number seems to be incorrect "3,466,39"

Author Response

This study reports mainly on the outcome of a de novo virus assembly that revealed a novel 29 kb single stranded positive-sense RNA virus that is the likely etiological agent of  mass mortalities in the giant freshwater prawn, Macrobrachium rosenbergii.  

The paper is well written and data is clearly presented.  I do have one major concern.  The wider screening activity (sections 2.9 & 3.8) appears to be flawed as this virus in question is a RNA virus without a DNA life stage.  So PCR in itself won't work.  It is unclear from the methods section whether the samples used in the wider screen was in fact cDNA.  I'm hoping that this is just a misunderstanding, and if so this must be corrected or clarified.

>> Provided clarification in methods, results and discussion that screens for MrGV were carried out on cDNA.

Similarly, I recommend the authors use the term RT-PCR rather than PCR when reporting on the screening results.  

>> RT-PCR seems an incorrect term to use for this paper, as reverse transcription and PCR were carried out separately.

I noticed one minor typo: line 243 - this number seems to be incorrect "3,466,39"

>> Corrected type in line 243. "3,466,39" corrected to 3,466,390".